



# A vorticity-and-stability diagram as a means to study potential vorticity nonconservation

Gabriel Vollenweider[1], Elisa Spreitzer[1], and Sebastian Schemm[1]

[1]Institute for Atmospheric and Climate Science, ETH Zürich, Zürich, Switzerland

**Correspondence:** Sebastian Schemm (sebastian.schemm@env.ethz.ch)

**Abstract.** The study of atmospheric circulation from a potential vorticity (PV) perspective has advanced our mechanistic understanding of the development and propagation of weather systems. The formation of PV anomalies by nonconservative processes can provide additional insight into the diabatic-to-adiabatic coupling in the atmosphere. PV nonconservation can be driven by changes in static stability, vorticity or a combination of both. For example, in the presence of localized latent heating,

the static stability increases below the level of maximum heating and decreases above this level. However, the vorticity changes in response to the changes in static stability (and vice versa), making it difficult to disentangle stability from vorticity-driven PV changes. Further diabatic processes, such as friction or turbulent momentum mixing, result in momentum-driven, and hence vorticity-driven, PV changes in the absence of moist diabatic processes. In this study, a vorticity-and-stability diagram is introduced as a means to study and identify periods of stability- and vorticity-driven changes in PV. Potential insights and

limitations from such a hyperbolic diagram are investigated based on three case studies. The first case is an idealized warm conveyor belt (WCB) in a baroclinic channel simulation. The simulation allows only condensation and evaporation. In this idealized case, PV along the WCB is first conserved, while stability decreases and vorticity increases as the air parcels move poleward near the surface in the cyclone warm sector. The subsequent PV modification and increase during the strong WCB ascent is, at low levels, dominated by an increase in static stability. However, the following PV decrease at upper levels is due to

a decrease in absolute vorticity with only small changes in static stability. The vorticity decrease occurs first at a rate of $0.5f$ per hour and later decreases to approximately $0.25f$ per hour, while static stability is fairly well conserved throughout the period of PV reduction. One possible explanation for this observation is the combined influence of diabatic and adiabatic processes on vorticity and static stability. At upper levels, large-scale divergence ahead of the trough leads to a negative vorticity tendency and a positive static stability tendency. In a dry atmosphere, the two changes would occur in tandem to conserve PV. In the case

of additional diabatic heating in the mid troposphere, the positive static stability tendency caused by the dry dynamics appears to be offset by the diabatic tendency to reduce the static stability above the level of maximum heating. This combination of diabatically and adiabatically driven static stability changes leads to its conservation, while the adiabatically forced negative vorticity tendency continues. Hence, PV is not conserved and reduces along the upper branch of the WCB. Second, in a full-fledged real case study with the Integrated Forecasting System (IFS), the PV changes along the WCB appear to be dominated

by vorticity changes throughout the flow of the air. However, accumulated PV tendencies are dominated by latent heat release from the large-scale cloud and convection schemes, which mainly produce temperature tendencies. The absolute vorticity decrease during the period of PV reduction lasts for several hours, and is first in the order of $0.5f$ per hour and later decreases





to $0.1f$ per hour when latent heat release becomes small, while static stability reduces moderately. PV and absolute vorticity turn negative after several hours. In a third case study of an air parcel impinging on the warm front of an extratropical cyclone,

changes in the horizontal PV components dominate the total PV change along the flow and thereby violate a key approximation of the two-dimensional vorticity-and-stability diagram. In such a situation where the PV change cannot be approximated by its vertical component, a higher-dimensional vorticity-and-stability diagram is required. Nevertheless, the vorticity-and-stability diagram can provide supplementary insights into the nature of diabatic PV changes.

## 1 Introduction

Since the seminal work of Hoskins et al. (1985), potential vorticity (PV) has proven to be a powerful quantity for the study of dry and moist atmospheric dynamics. Under adiabatic and frictionless conditions, PV is materially conserved (Ertel, 1942). Given suitable balanced conditions, the current state of the atmosphere is, in terms of its pressure, temperature and wind distributions, fully described by the PV field (Davis and Emanuel, 1991; Davis, 1992; Stoelinga, 1996), which makes "PV thinking" an attractive perspective for the study of atmospheric dynamics.

Many atmospheric systems can be understood in terms of their PV distribution. On the planetary scale, the PV framework can be used to study, for example, the temporal evolution of the Rossby wave structure and the upper-level jet stream (e.g. Davies and Rossa, 1998; Grams et al., 2011; Schemm et al., 2013; Hoskins, 2015; Teubler and Riemer, 2016; Joos and Forbes, 2016; Harvey et al., 2020). On the synoptic scale, extratropical cyclones are of particular interest, since they redistribute momentum, heat, and moisture, and have high impacts on the weather conditions (e.g. Held, 1975; Chang et al., 2002). Mature cyclones

are typically associated with specific PV signatures, namely, an upper-level positive PV anomaly (corresponding to a trough), and a mid-level positive PV anomaly related to diabatic processes plus a surface PV anomaly corresponding to the cyclone warm sector (Davis and Emanuel, 1991; Reed et al., 1992; Čampa and Wernli, 2012). The accompanying fronts are often characterized by elongated bands of positive PV (e.g., Fehlmann and Davies, 1999; Lackmann, 2002; Schemm and Sprenger, 2015; Attinger et al., 2019). The PV framework has also been used to examine mesoscale phenomena such as mesoscale

dynamics along fronts (e.g. Parker and Thorpe, 1995; Fehlmann and Davies, 1999; Lackmann, 2002; Igel and van den Heever, 2014) or mesoscale convection (e.g. Chagnon and Gray, 2009; Clarke et al., 2019; Oertel et al., 2020). Particularly on the synoptic scale and below, the atmospheric circulation is significantly affected by heating and cooling due to diabatic processes and the PV framework has intensively been used to study diabatic PV modification on these scales (e.g. Stoelinga, 1996; Chagnon et al., 2013; Schemm and Wernli, 2014; Büeler and Pfahl, 2017; Attinger et al., 2019; Spreitzer et al., 2019).

Diabatic processes that alter PV can broadly be divided into two categories: (i) nonconservative forces (e.g., friction, turbulent mixing), which cause momentum tendencies, and (ii) diabatic processes (e.g., phase changes, radiative transfer), which cause potential temperature tendencies. Some processes cause momentum and potential temperature tendencies (e.g., convection, turbulence). Momentum tendencies affect the (absolute) vorticity ($\boldsymbol{\omega}_{a}$), and potential temperature tendencies affect the local gradient of potential temperature ($\boldsymbol{\nabla}\theta$), and it is their combined nonlinear effect that causes a change in PV. A diabatic

process initially cause changes in potential temperature, but later on, this perturbation is converted to a change in the vortic-





ity (cf. the concept of "latent vorticity" generation in Chagnon and Gray, 2009) also when the atmosphere adjusts to a new balanced state in a process of hydrostatic-geostrophic adjustment (Blumen, 1972; Chagnon and Bannon, 2001; Egger, 2009) during which the inertia, gravity, and sound waves radiate away from the heating perturbation (Bretherton and Smolarkiewicz, 1989; Chagnon and Bannon, 2001). It remains difficult to assess whether PV changes are primarily driven by vorticity or stabil-

ity changes and whether it is possible to disentangle them since it can be a nonlinear combination of changes in both quantities that result in a PV change.

To assess which nonconservative process causes a change in absolute vorticity along the flow of an air parcel, and which one modifies the gradient of potential temperature, we explore PV changes along the flow of air parcels in a two-dimensional phase portrait, splitting PV changes into a vorticity component and a static stability component. Examining the phase space

of Lagrangian PV changes offers the possibility to assess whether and when it is possible to split the temporal evolution of PV along a parcel trajectory in changes that are vorticity- or static stability-driven. In combination with the accumulated PV changes due to the different physical processes, it can then be determined whether or not there are any adjustments from perturbed vorticity to modified stability (or vice versa) along the trajectory.

In the following sections, we first introduce in Section 2 the methodology to obtain the vorticity-and-stability diagram and

the simulations conducted to study the Lagrangian PV evolution in idealized and real-case extratropical cyclones.

## 2 Methodology

### 2.1 Lagrangian PV framework

The material evolution of PV along an air parcel trajectory is given by

$$\frac{\mathrm{D}}{\mathrm{D}t}\mathrm{PV} = \frac{1}{\rho}\left(\mathbf{rot}\left(\boldsymbol{F}\right)\cdot\boldsymbol{\nabla}\theta + \boldsymbol{\omega}_{\mathrm{a}}\cdot\boldsymbol{\nabla}\left(\frac{\mathrm{D}\theta}{\mathrm{D}t}\right)\right) \tag{1}$$

where the operator $\mathrm{D}/\mathrm{D}t$ denotes the material derivative, $\mathbf{rot}$ is the rotation operator, $\boldsymbol{F}$ represents the sum of all nonconservative forces (momentum tendencies), $\mathrm{D}\theta/\mathrm{D}t$ stands for the combined effect of all potential temperature tendencies and PV is defined as

$$\mathrm{PV} = \frac{\boldsymbol{\omega}_{\mathrm{a}}\cdot\boldsymbol{\nabla}\theta}{\rho}, \tag{2}$$

where $\boldsymbol{\omega}_{\mathrm{a}} = 2\boldsymbol{\Omega} + \mathbf{rot}(\boldsymbol{u})$ is the absolute vorticity, $\theta$ is the potential temperature, and $\rho$ is the density of air. Equation (1) shows

that for a frictionless ($\boldsymbol{F} = \mathbf{0}$) and adiabatic ($\mathrm{D}\theta/\mathrm{D}t = 0$) flow, PV is materially conserved. Therefore, any changes in the PV field can be attributed to either nonconservative forces or moist diabatic processes. Considering that equation (1) describes the changes in PV following the atmospheric motion, we adopt the Lagrangian perspective in this study. Air parcel trajectories are calculated using the Lagrangian Analysis Tool LAGRANTO (Wernli and Davies, 1997; Sprenger and Wernli, 2015) and all variables appearing in equation (1) are traced along the trajectories. This allows us to compute the corresponding PV tendencies

along the trajectories. The PV tendencies are split into contributions from various nonconservative process as outlined in the





following section. The individual PV tendencies are also integrated along the trajectory to yield the accumulated PV (APV; cf. Spreitzer et al., 2019). APV is a convenient measure to quantify the evolution of PV relative to the starting point of the trajectory and the fractional contribution of different processes to the total Lagrangian PV change in a given time interval. The integral of the PV tendencies along a parcel trajectory for each process ($\chi$; $\mathrm{PVR}_\chi$) is approximated by the following sum for forward trajectories:

$$\mathrm{APV}_\chi(t_k) \approx \sum_{p=1}^{k} \mathrm{PVR}_\chi(t_{p-1})\, \Delta t, \text{ where } k \in \{0,\dots,n\}, \tag{3}$$

where $\Delta t$ is the corresponding model output time step (hourly in this study).

## 2.2 The vorticity-and-stability diagram

In many but not all large-scale flow situations, PV is well approximated by its vertical component ($\mathrm{PV}_\mathrm{ver}$), given by $\mathrm{PV}_\mathrm{ver} = \rho^{-1} \cdot \zeta_\mathrm{a} \cdot \partial_z\theta$, where $\zeta_\mathrm{a}$ is the vertical component of the absolute vorticity ($\boldsymbol{\omega}_\mathrm{a}$). Thus, diabatic PV changes result from combined vorticity-and-stability changes, which we aim to understand using the vorticity-stability diagram. PV, $\mathrm{PV}_\mathrm{ver}$, the absolute vorticity, and static stability components are traced independently along air parcel trajectories and the latter two are used as the coordinates of a two-dimensional phase space. The temporal evolution of PV along a Lagrangian air parcel trajectory can then be displayed as a phase space trajectory in this diagram. Below, we discuss an illustrative example of this diagram.

Figure 1 shows a schematic illustration of the vorticity-and-stability diagram, with static stability on the vertical axis and absolute vertical vorticity on the horizontal axis. Lines of constant PV are shown as red hyperbolas. Rotated hyperbolas (gray) that are perpendicular to the lines of constant PV are also shown. These indicate the maximum possible changes of PV at every location in the phase space. What matters most in understanding the drivers of PV changes is the orientation of the rotated hyperbolic grid (gray) relative to the horizontal and vertical axes. PV changes in regions where the gray hyperbolas are oriented more vertically than horizontally tend to be driven by changes in the static stability. These are indicated by blue dots. PV changes in regions where the rotated gray hyperbolas are oriented more horizontally than vertically tend to be vorticity-driven PV changes (indicated by red dots). Due to the hyperbolic nature of the PV isoline, changes in static stability on the right and left side of the diagram translate into a stronger PV change compared to a similar change in static stability that occurs in the upper and lower side of the diagram. The opposite is true for absolute vorticity changes.

Each colored dot in the diagram represents a time step along an air parcel trajectory. The first time step is labeled as $t = t_0$, and the last one corresponds to $t = t_n$. PV at the first and the last time step are 3 PVU, respectively, but result from different combinations of vorticity and static stability. During the initial phase, the PV change along the trajectory is driven by changes in the static stability (blue dots). Later, in the upper left corner of the diagram, the PV changes are vorticity-driven (red dots). To obtain an approximate measure for the relative contributions from vorticity-and-stability changes, the PV change is linearized and expressed as the sum of changes due to vorticity-and-stability. In pressure coordinates, the vertical PV is given by (in a simplified form), $\mathrm{PV}_\mathrm{ver} = -g \cdot \zeta_{\mathrm{a},p} \cdot \partial_p\theta$, where the subscript $p$ indicates that the vertical component of (relative) vorticity is



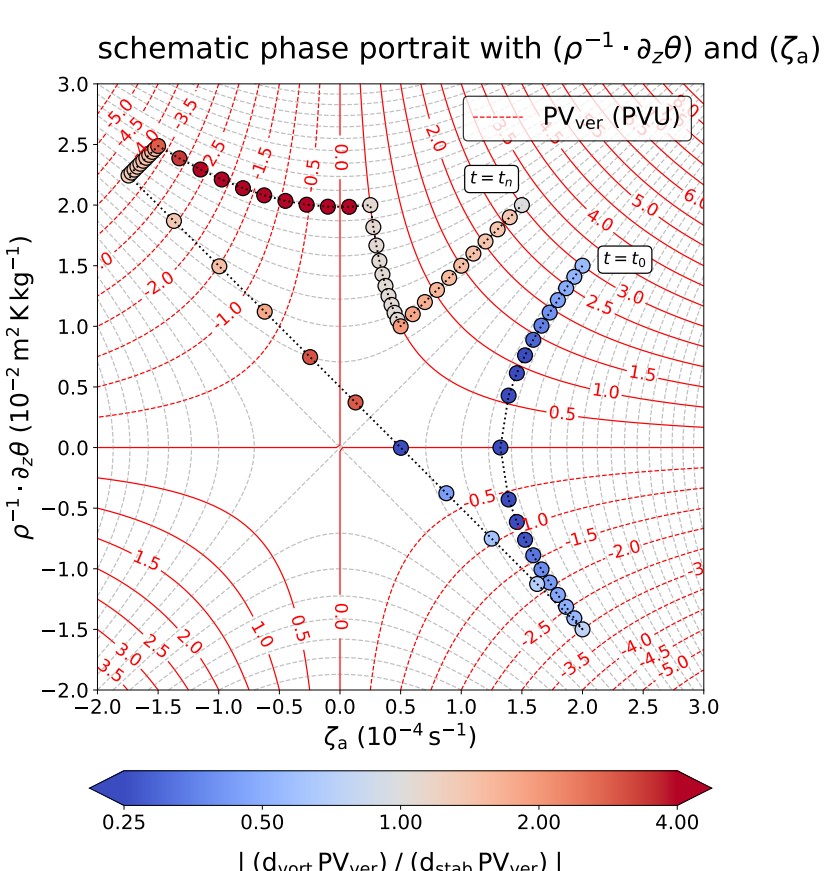

**Figure 1.** The vorticity-and-stability diagram for the rate of change of the vertical PV ($\mathrm{PV_{ver}}$) component. The horizontal axis shows the absolute vorticity and the vertical axis shows the static stability component of $\mathrm{PV_{ver}}$. Lines of constant $\mathrm{PV_{ver}}$ are indicated by red hyperbolas. A second rotated hyperbolic grid (dashed gray lines) is also included. This grid is perpendicular to the lines of constant $\mathrm{PV_{ver}}$ and indicates the directions of greatest PV changes. Additionally shown is an idealized trajectory through the diagram (starting at $t = t_0$ and ending at $t = t_n$), which is colored according to the ratio of the PV change in absolute vertical vorticity relative to static stability. Red dots indicate that absolute vertical vorticity dominate the change in $\mathrm{PV_{ver}}$, while blue indicates that static stability dominates. To obtain the ratio, linearization is used (see text for more details).

calculated on a surface of constant pressure. Using this definition of vertical PV, we obtain:

$$\delta \, \mathrm{PV_{ver}} \approx \underbrace{\delta \, \zeta_{\mathrm{a},p} \cdot (-g \cdot \partial_p \theta)}_{\delta_{\mathrm{vort}} \, \mathrm{PV_{ver}}} + \underbrace{\zeta_{\mathrm{a},p} \cdot \delta \, (-g \cdot \partial_p \theta)}_{\delta_{\mathrm{stab}} \, \mathrm{PV_{ver}}} . \tag{4}$$





The ratio of these two components (in the absolute value) is then used to color the change in PV (red and blue points in
Figure 1) at each time step.

The vorticity-and-stability diagram is based on the vertical component of PV only. Therefore, it is important that $PV =$
$PV_{ver}$ holds to a good approximation for the diagram to be meaningful. If the approximation is not satisfied, any changes in the
full PV can also be associated with variations in the horizontal PV, requiring a higher-dimensional phase space. Additionally,
the vorticity-and-stability diagram of $PV_{ver}$ may be misleading since a change in the vertical PV might also be due to exchanges
between horizontal PV and vertical PV. From our experience, the approximation breaks down typically near fronts, in the
presence of strong horizontal wind shear and/or strong horizontal gradients of potential temperature. In that case, the horizontal
PV ($PV_{hor} = PV - PV_{ver}$) must be taken into account, making the phase space of PV six-dimensional (three absolute vorticity
coordinates and three coordinates for the gradient of potential temperature). We therefore trace the full three-dimensional Ertel
PV along the parcel trajectories to assess how well PV is approximated by its vertical component.

**2.3   Simulations**

To study Lagrangian PV changes through the lens of the vorticity-and-stability diagram, we investigate PV along trajectories
from three different case studies associated with extratropical cyclones.

First, we study an idealized warm conveyor belt: The first case (section 3.1) is based on idealized baroclinic channel simu-
lations of Schemm et al. (2013). Here, we use the original data of Schemm et al. (2013), which were archived at ETH Zurich.
The simulations are performed with the regional numerical weather prediction model COSMO on an $f$-plane with a horizontal
grid spacing of approximately 21 km. The simulations were originally used to study warm (WCB) and cold conveyor belts
(CCB) and their influence on the downstream and cyclone developments in Schemm et al. (2013) and Schemm and Wernli
(2014). Diabatic processes in these simulations are limited to condensation and evaporation via a simple saturation adjustment
and nonconservative forces are limited to numerical and turbulent diffusion. The setup of the simulations is further introduced
in detail in Schemm et al. (2013).

Second, two real-cases are taken from a dataset by Spreitzer et al. (2019), who employed the integrated forecasting system
(IFS) from the European Centre for Medium-Range Weather Forecasts (ECMWF) to simulate a North Atlantic extratropical
cyclone. The version of the IFS model used for the simulation, cycle 43R1, was operational at the ECMWF from November
2016 to July 2017. The model is hydrostatic, semi-implicit, and semi-Lagrangian. The forecasts were run at a cubic spectral
truncation of TCo639, which translates to a global average horizontal grid spacing of 18 km. The model output was archived
every hour and horizontally interpolated to a regular grid with a resolution of 0.25°. The sub-grid scale temperature and
momentum tendencies include clouds and large-scale precipitation (Forbes and Tompkins, 2011; Forbes and Ahlgrimm, 2014),
moist convection (Tiedtke, 1989; Bechtold et al., 2008), long-wave and short-wave radiation (Mlawer et al., 1997; Iacono et al.,
2008), and sub-grid turbulent transport (Lott and Miller, 1997; Beljaars et al., 2004; Siebesma et al., 2007; Orr et al., 2010;
Köhler et al., 2011). These parametrization schemes generate sub-grid tendencies of temperature and/or momentum, leading to
material changes of PV. Note that in the IFS sub-grid momentum tendencies only exist for the horizontal wind components $u$





and $v$. For further details, see Spreitzer et al. (2019) and ECMWF (2016). The following tendencies were archived every hour from the model run:

- temperature tendencies from the clouds and large-scale precipitation: $(\mathrm{D}T/\mathrm{D}t)_{\mathrm{ls}}$,

- temperature and momentum tendencies from convection: $(\mathrm{D}u/\mathrm{D}t)_{\mathrm{conv}}, (\mathrm{D}v/\mathrm{D}t)_{\mathrm{conv}}, (\mathrm{D}T/\mathrm{D}t)_{\mathrm{conv}}$,

- temperature tendencies from the long-wave radiation: $(\mathrm{D}T/\mathrm{D}t)_{\mathrm{lw}}$,

- temperature tendencies from the short-wave radiation: $(\mathrm{D}T/\mathrm{D}t)_{\mathrm{sw}}$,

- temperature and momentum tendencies from turbulence: $(\mathrm{D}u/\mathrm{D}t)_{\mathrm{turb}}, (\mathrm{D}v/\mathrm{D}t)_{\mathrm{turb}}, (\mathrm{D}T/\mathrm{D}t)_{\mathrm{turb}}$.

These tendencies are used to compute the PV tendencies according to equation (1) and APV according to equation (3) for
the cyclone's WCB and a set of parcel trajectories that impinge on the warm front.

## 3 The vorticity-and-stability diagram in practice

### 3.1 Idealized WCB trajectories

In the idealized baroclinic wave simulation of Schemm et al. (2013), a cyclone forms three days after the start of the baroclinic channel simulation. At day five, the primary cyclone is well developed and accompanied by an evolving ridge downstream
and well-marked surface fronts (Fig. 2). Figure 2a displays a vertical cross section which transects the northern part of the ridge where a WCB outflow is located in the upper-tropospheric low-PV region above 6 km seen in Fig. 2a (cf. Fig. 6 in Schemm et al., 2013). Schemm et al. (2013) computed 48 h forward air parcel trajectories starting at low levels on day three and selected those that ascended at least 600 hPa in 48 h$^{-1}$. This ascent criterion is well established to identify the WCB air masses associated with the cyclone (Wernli and Davies, 1997). They form a coherent air stream, which is then averaged to
obtain one representative mean trajectory. The pathway of the mean WCB trajectory in hourly steps and colored by height is shown in Fig. 3c. For animations of this WCB, see the supplementary videos in Schemm et al. (2013) and Schemm and Wernli (2014).

The temporal PV evolution is typical of a WCB. Initially, the air remains at low levels ($z < 0.5$ km) with constant PV values (Fig. 3a). Then, at approximately 28 h after the start of the trajectories, the air begins to ascend until it reaches the upper
troposphere at approximately 7.5 km altitude. The ascending WCB is associated a strong diabatic heating rate from cloud condensation (Fig. 3b) and a corresponding initial increase in PV of roughly 1 PVU followed by a subsequent decrease of about 1.25 PVU (Fig. 3a). This agrees well with the general framework of WBCs, with a PV increase below a maximum of diabatic heating, and PV decrease above this maximum (Eliassen and Kleinschmidt, 1957; Wernli and Davies, 1997). As indicated in Fig. 3a, the $\mathrm{PV}_{\mathrm{ver}}$ is a reasonable approximation for the full PV throughout the 48 hour period.
Since PV is, in general, well approximated by $\mathrm{PV}_{\mathrm{ver}}$, the two-dimensional vorticity-and-stability diagram is representative for the evolution of the full PV (Fig. 3a). For the first 30 h of the trajectory, PV is well conserved along the flow but static

**Figure 2.** Synoptic situation and vertical cross sections five days after the start of the idealized baroclinic channel simulation. The first vertical cross section (a) shows the full PV (shading; PVU) and potential temperature (red contours; in steps of 5 K). The second vertical cross section (c) shows the difference between the full PV and the vertical PV component. Also shown are (b) upper-level PV (shading; PVU) on 320 K together with sea level pressure (blue contours; in steps of 15 hPa) and (d) surface temperature together with sea level pressure (red contours; in steps of 15 hPa).

stability ($\rho^{-1} \cdot \partial_z \theta$) decreases, while absolute vertical vorticity ($\zeta_a$) increases and the trajectory runs parallel to a red hyperbola in the vorticity-and-stability diagram (Fig. 3d). After this initial phase of PV conservation, the air parcels approach the front, and we observe a mild PV increase for one time steps, which is driven by a change in the vorticity component of PV (single dark red point in Fig. 3d after approximately 30 h), eventually related to strong tilting of the isentropic surface and of the vorticity vector. Next, however, PV starts to change vigorously at $t = 29\,h$ (Fig. 3a), and the blue dots in Fig. 3d indicate that





**Figure 3.** (a) Temporal evolution of PV and $PV_{ver}$ along the cyclone's WCB. (b) Corresponding evolution of diabatic heating due to condensation and evaporation. (c) The hourly evolution of the WCB colored by the mean air parcel height, sea-level pressure (blue contour; in steps of 15 hPa) and the 2 PVU contour (black) on 320 K. (d) The vocticity-and-stability diagram and the temporal evolution of the mean parcel trajectory. The points along the trajectory are colored according to the importance of modified absolute vertical vorticity relative to changes in static stability, both fields are traced along every WCB trajectory.





the PV increase is now driven by a change in the static stability, which agrees with what can be expected from the increase in static stability below the height of maximum condensation (Fig. 3b). To conserve PV, absolute vorticity would now need to decrease while static stability increase, but this is not the case and both increase for a period of $5--6$ hours until PV reaches it

maximum value at $t = 34\,h$ (Fig. 3a), which notably is before the maximum in latent heating is observed at $t = 37\,h$ (Fig. 3b). The maximum potential temperature heating rate at $t = 37\,h$ is close to $2.5\,\mathrm{K\,h^{-1}}$. The PV decrease from $t = 34\,h$ onward is dominated by a modification of absolute vertical vorticity (as indicated by the red dots in Fig. 3d) with only little to no change in static stability. Absolute vorticity and PV decrease simultaneously from $t = 34\,h$ onward and over the next 12 hours until the vertical PV becomes slightly negative at $t = 46\,h$. The prolonged decrease in absolute vorticity first occurs a rate of

$0.5\,f^{-1}\,h^{-1}$ under strong latent heat release for a period of $4--5$ hours. Afterward and when latent heat release passed its maximum the reduction rate of absolute vorticity weakens and occurs at a rate of $0.2\,f^{-1}$ per hour.

The change in sign of the absolute vertical vorticity is the reason for the negative values of $\mathrm{PV_{ver}}$, which is observed during the last part of the trajectory when absolute vorticity turns negative (Note, the simulations are performed on a $f$ plane with constant $f = 10^{-4}$. Hence relative vorticity turns already negative at $t = 38\,h$ in Fig. 3d). Negative absolute vorticity is

consistent with the fact that negative PV values are usually associated with inertial instabilities, $\zeta_\mathrm{a} < 0$, as opposed to static instabilities, corresponding to $\rho^{-1}\cdot\partial_z\theta < 0$. However, note that the total PV is not negative (Fig. 3a), indicating that the negative vertical PV component is balanced by horizontal PV components such that it yields zero total PV.

### 3.1.1 Mechanisms that underlie the observed temporal evolutions of stability and vorticity

In the previous paragraph, we identified vorticity- and stability-driven periods of PV nonconservation. A detailed analysis of

tendency equations for vorticity and stability along the idealized WCB are of merit but beyond the scope of this study. For example, for the stability analysis, it would be insightful to study the vertical component of the frontogenesis equation,

$$\frac{\partial}{\partial t}\frac{\partial \theta}{\partial z} = -\mathbf{u}\cdot\nabla\frac{\partial \theta}{\partial z} + \frac{\partial}{\partial z}\dot{\theta} - \left(\frac{\partial u}{\partial z}\frac{\partial \theta}{\partial x} + \frac{\partial v}{\partial z}\frac{\partial \theta}{\partial y} + \frac{\partial w}{\partial z}\frac{\partial \theta}{\partial z}\right), \tag{5}$$

alongside the vorticity equation. In this paragraph, we speculate how different diabatic and adiabatic mechanisms contribute to the observed vorticity and stability changes.

Consider an air parcel bounded above and below by two isentropic surfaces and the presence of a heating rate gradient. We know that a heating rate gradient exists because PV is not conserved (Eq. 1 and $\mathbf{F} = 0$). When the air parcel is above the maximum latent heat release, the downward displacement of the isentropic surface at the parcel bottom is larger compared to that of the isentropic lid, because the former is located closer to the maximum in latent heat release. As a result, we expect column stretching, reduced static stability and vorticity generation. However, what is observed along the WCB is a nearly

constant static stability and an increase in vorticity. In the upper troposphere, the WCB air parcel is located in a region ahead of the upper-level trough, which is characterised by large-scale ascent and divergence. The large-scale divergence ahead of the trough drives a reduction in vorticity and stronger divergence aloft also drives a column shrinking. In a dry atmosphere the two changes would occur in tandem such that they conserve PV. In dry baroclinic channel simulations, the static stability





indeed increases in the WCB outflow near the tropopause (Kunkel et al., 2016). But the diabatic influence counter-act the
adiabatically-driven tendency to increase the static stability. Together, the diabatic and adiabatic influence on the static stability
seem to balance each other in the upper troposphere, while the large-scale divergence still forces a vorticity reduction. Hence,
PV is not conserved and reduces.

   The above presented view is of course not complete. The vorticity vector likely tilts in the vicinity of the upper-level front
and the vorticity reduction could be re-enforced by vorticity tilting. It could also be that the vorticity decrease is related to
the tendency of the atmosphere to adjust to a new balanced state while emitting gravity waves. The vorticity reduction occurs
at a rate of $0.2 - 0.5\,f^{-1}\,h^{-1}$. However, negative absolute vorticity, which indicates inertia instability, is unlikely the desired
outcome of an geostrophic adjustment process. Next, we explore a similar WCB development but this time using a real-case
simulation using the IFS.

### 3.2   Real-case WCB trajectories

The examined extratropical cyclone had its genesis off the US East Coast on 4 February 2017 and developed into a mature
storm as it crossed the North Atlantic. On 6 February 2017, the cyclones was located south of Greenland (Fig. 4). The cyclone's
WCB trajectories were examined in this section, but we restrict our analysis to those that are affected by negative PV in the
WCB outflow, because negative PV generation has lately received enhanced attention. The WCB parcel trajectories are thus
started from a region of absolute negative PV in the upper-level ridge on 12 UTC 6 February 2017 and are computed 24 h
backwards. The starting location is indicated in the W–E cross section at 300 hPa near 4°E (white box in Fig. 4a). The stage in
the cyclone life cycle is in good agreement with the previously examined idealized cyclone as can be seen when comparing for
example the two cross sections Fig. 4a and Fig. 2a. The horizontal PV component in the vicinity of the WCB outflow is close
to zero (Fig. 4c), which is a good first indicator that the vertical PV component and its evolution are representative of the full
PV evolution along this WCB.

Figure 5a shows that PV is indeed well approximated by its vertical component along the corresponding WCB trajectories.
Again, we average across the WCB trajectory sample. The air parcels start at low levels ($p > 700$ hPa) and end in the upper
troposphere ($p \approx 350$ hPa). During the ascent, there is an initial increase in PV of approximately 0.5 PVU followed by
a subsequent decrease of about 1 PVU (Fig. 5c). Figure 5b, which shows the accumulated PV due to different processes,
indicates that the evolution of PV is again linked diabatic heating due to the large-scale cloud scheme (e.g., condensation) and
convection (which are both added into one diabatic heating rate for simplicity). In agreement with the general framework of
WCBs, PV increases below the maximum of diabatic heating, and decreases above this maximum (Eliassen and Kleinschmidt,
1957; Wernli and Davies, 1997). In contrast to the idealized case, both the vertical and full PV change sign in the WCB outflow
and are negative during the last 7–6 hours of the parcel trajectory (Fig. 5a). Negative PV can be associated with static instability,
inertial instability, or symmetric instability (Hoskins, 2015). It is assumed that static instability is released on a time-scale of
a few minutes, inertial instability can however last for several hours (Schultz and Schumacher, 1999; Thompson et al., 2018),
making long-lasting negative PV anomalies possible. Indeed, the negative PV values in the WCB outflow result from a change

**Figure 4.** Similar to Figure 2 but for a real cyclone over the eastern North Atlantic at 12 UTC 06 February 2017.

of sign of absolute vorticity. Thus, the long-lasting negative PV values are found to be associated with inertial instabilities where: $\zeta_{\mathrm{a},p} < 0$, rather than static instabilities.

Because the temporal evolution of PV is reasonably well approximated by its vertical component $\mathrm{PV_{ver}}$, the two-dimensional
vorticity-and-stability diagram is representative of the full PV evolution (Fig. 5d). It shows that the evolution in PV is mostly dominated by changes in absolute vertical vorticity rather than modifications of static stability. It is only during the change from PV increase to PV decrease that static stability changes become more important for a short time period when both make almost equal contribution for 3-4 hours. However, as PV continues to decrease, the absolute vorticity changes increasingly dominate the PV change again, consistent with the fact that the phase space trajectory moves into a region of the diagram where the
rotated gray hyperbolas are oriented more horizontally than vertically. Nevertheless, static stability decreases as well, which

**Figure 5.** Similar to Figure 3 but for the starting region specified in Section 3.2 at 12 UTC 06 February 2017. The parametrizations of the IFS simulations include more nonconservative processes compared with the idealized case and accumulated PV is shown for diabatic tendencies due to large-scale clouds (ls) plus convection (conv), long-wave radiation (lw), short-wave radiation (sw), and turbulence (turb).

indicates column stretching. As discussed in Section 3.1.1, large-scale divergence in the WCB outflow could be the primary mechanism driving the reduction in vorticity and – in principal – it drives an increase in static stability. However, the heating



(a) PV and $\theta$

(b) isentropic PV and $p_{msl}$

(c) PV$_{hor}$ and $\theta$

(d) T and $p_{msl}$

**Figure 6.** Same as Figure 4 but for the region specified in section 3.3 at 00 UTC 06 February 2017.

rate gradient and other diabatic mechanism weaken the adiabatic influence on the static stability. Here, it seems as if they cause a stability reduction. Hence, the diabatic and adiabatic interplay causes PV nonconservation, specifically, a PV reduction in the

WCB outflow above the heating rate gradient, driven by vorticity reduction from the large-scale divergence ahead of the trough and diabatic heating counteracting the stability increase needed to conserve PV.

## 3.3   Real case: Warm front trajectories

For this third case, we intentionally chose a situation where the vertical PV component alone is not a good approximation for the temporal evolution of the full three-dimensional PV along a trajectory. Consideration is given to the same cyclone as in the





second case study, but at 00 UTC 6 February 2017. Figure 6b shows the upper-level PV distribution and the sea level pressure and the vertical cross section in Fig. 6a is located slightly north of the cyclone center and transects the bent-back front and the warm front, both associated with positive PV anomalies. Figure 6c shows that the horizontal PV components are nonnegligible near the front (blue shading near 25°E at 800 hPa). The 24 h-backward trajectories are started in a positive PV anomaly ( > 2PVU) within this transect at the front near $20 - -25$°E at 800–750 hPa (start box is shown in Fig. 6a and c), at 00 UTC 6 February 2017. The starting positions of the backward parcel trajectories in this target region are selected according to the following criteria:

$$\lambda = (-23 \text{ to } -21.5)\,°\text{E}, \tag{6a}$$

$$\varphi = (54.5 \text{ to } 55.5)\,°\text{N}, \tag{6b}$$

$$p = (725 \text{ to } 775)\,\text{hPa}, \tag{6c}$$

$$\text{PV}(t_{\text{ini}}) \geq 2\,\text{PVU}. \tag{6d}$$

The selected trajectory bundle consists of 52 air parcel trajectories. In the following, the mean properties averaged over all trajectories are considered.

Figure 6c shows that the temporal evolution of PV is not always approximated well by the vertical component because the horizontal PV component is strong. It holds for the first two thirds of the trajectory, while they experience only a weak PV increase. But after time step $t = -8$ h (8 hours prior to arrival in the target region), the vertical PV starts to overestimate the actual PV. This is consistent with the fact that the last part of the trajectory is near the front, where the horizontal PV ($\text{PV}_{\text{hor}}$) takes negative values (blue shading in Fig. 6c). The strong increase in PV occurs as the air ascends near the front. At $t = -8$ h, the air is still in the boundary layer at pressure levels of about 925 hPa, and it reaches heights of approximately 750 hPa within the following eight hours. During this period horizontal PV generation is observed in the vicinity of mesoscale convection – as described in Chagnon and Gray (2009); Oertel et al. (2020) and Oertel and Schemm (2021).

Considering the first part of the trajectory, the phase portrait (Fig. 7d) shows that from the start ($t = -24$ h near the center of the diagram) and up to time step $t = -12$ h, there are only very slight changes in both static stability and absolute vertical vorticity and the weak increases in PV is dominated by modifications of static stability (blue points gathering in the center of the diagram). As seen in Fig. 7b from the accumulate diabatic PV, these changes are at first due to turbulence, but after $t = -13$ h, convection takes over as the main process. A closer inspection shows that the large-scale cloud scheme is negligible, so we observe the onset of convection, which is what we expect from the case studies in Oertel et al. (2020); Oertel and Schemm (2021) and the formation of horizontal PV. From $t = -12$ h to $t = -8$ h, this translates into a PV increase dominated by an increase in the absolute vertical vorticity (red points in Fig. 7d).

For the last eight hours of the trajectory, the evolution of the vertical PV is again dominated by changes in static stability. This is consistent with the very dominant role of both large-scale clouds and convection from $t = -5$ h onward to $t = 0$ h. The associated temperature tendencies lead to increases in static stability (convective momentum tendencies are small). In particular, there is a strong increase in diabatic heating towards the end of the trajectory (blue line in Figure 7d). This heating is now attributed to the large-scale cloud scheme and is no longer due to convection. Thus, the trajectory rises and approaches



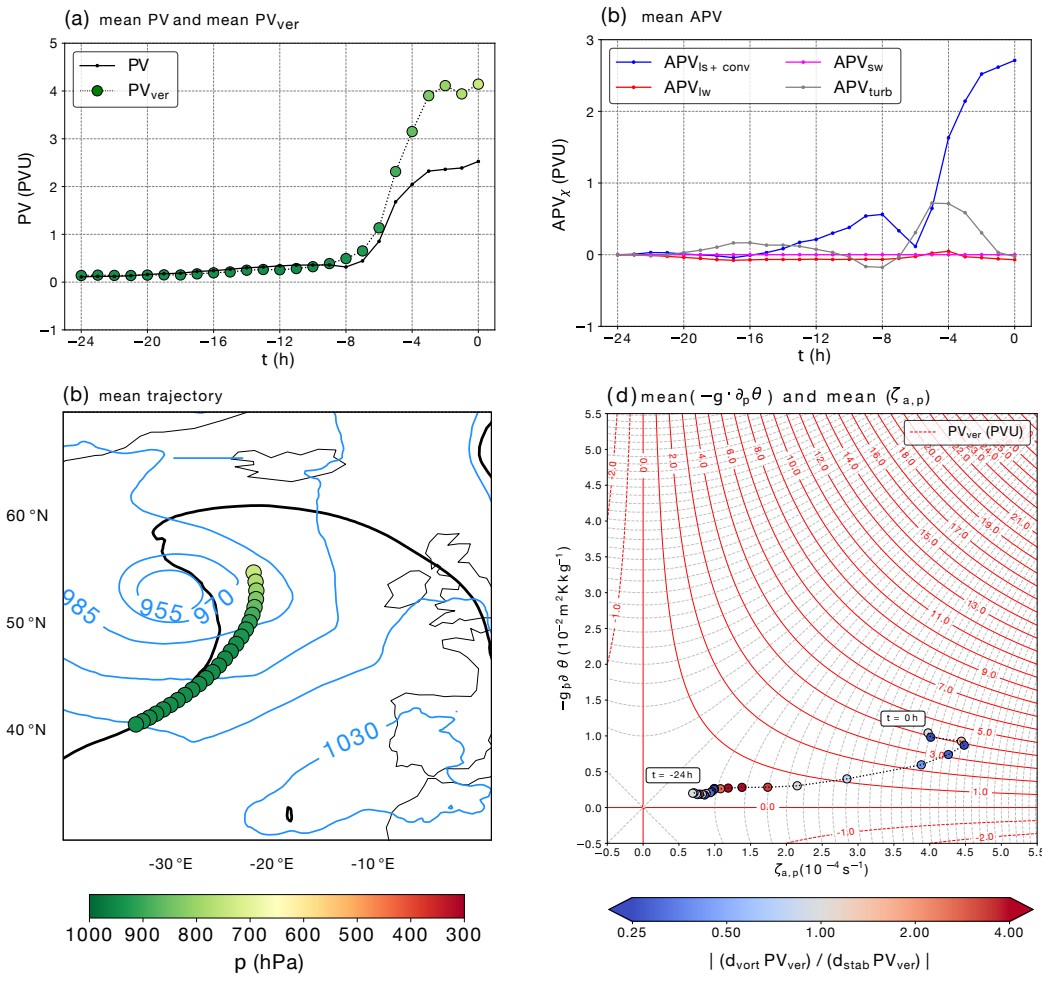

**Figure 7.** Same as Figure 5 but for the starting region specified in section 3.3 at 00 UTC 06 February 2017.

a maximum of diabatic heating. During the last four time steps, the vertical PV follows the 4 PVU isoline. This is the part of
the trajectory where the absolute vertical vorticity decreases (Fig. 7d). For these final time steps, turbulence is another process
leading to an increase in PV.

## 4 Conclusions

The study of PV nonconservation provides insight into the diabatic to adiabatic coupling in the atmosphere. The PV framework
provides a coherent framework to study how cloud-related diabatic processes, as well as various other nonconserving processes,
affect the temporal evolution of the atmosphere. Diabatic processes cause temperature and momentum tendencies, which
translate into stability- or vorticity-driven PV changes, respectively. However, the nonlinear nature of PV makes it difficult to





disentangle stability from vorticity-driven changes. In this study, we explore the usefulness of displaying the Lagrangian PV evolution along air parcel trajectories in a two-dimensional vorticity-and-stability diagram in which the lines of constant PV are depicted as hyperbolic curves. The vorticity-and-stability diagram is two-dimensional as long as PV is reasonably well
approximated by the vertical components of vorticity and potential temperature gradient ($\mathrm{PV} \approx \mathrm{PV}_{\mathrm{ver}} = -g \cdot \zeta_{\mathrm{a},p} \cdot \partial_p \theta$).

Three case studies are performed, an idealized WCB and a real-case WCB. The idealized baroclinic channel simulations allows only for condensation and evaporation, while the real-case simulation with the full-fledged IFS contains temperature and momentum tendencies from the large-scale cloud scheme, convection, radiation and turbulence. The temporal evolution of PV along WCB air parcel trajectories, with the characteristic PV increase at low levels and a decrease at upper levels, is in both
simulations reasonably approximated by $\mathrm{PV}_{\mathrm{ver}} = -g \cdot \zeta_{\mathrm{a},p} \cdot \partial_p \theta$. In the idealized simulation, it is found that PV along the WCB is first conserved, while vorticity already increases and stability decreases as the air parcels move poleward in the cyclone's warm sector. The subsequent phase of PV nonconservation (PV increase) at low levels is driven by an increase in static stability while vorticity continues to increase as well. Static stability below the maximum in condensation is expected to increase because the strong heating above brings the isentropic surfaces closer together. At upper levels, however, the reduction in PV
is driven by changes in absolute vorticity and the decrease in vorticity already starts before we observe the maximum in latent heat release along the WCB trajectories. Because the idealized simulation allows only for diabatic temperature tendencies, the dominance of absolute vorticity-driven PV change in the WCB outflow does not result from diabatic momentum modification. There are thus other mechanisms at play but a complete analysis of the temporal evolution of tendency equations for vorticity and stability along the WCB is beyond the goal of this study and we here provide only some discussion points.

Static stability above the maximum in latent heat release will decrease, because the isentropic bottom of an air parcel, which is bounded between two isentropic surfaces, is stronger deflected downward compared with the isentropic lid. We would thus expect column stretching and vorticity generation. Large-scale divergence ahead of the trough in the WCB outflow will however impose strong adiabatic tendencies for vorticity reduction and stability increase near the tropopause (as seen in idealized dry simulations, e.g., Kunkel et al. (2016)). In the absence of diabatic processes, this vorticity reduction and stability increase (i.e,
column shrinking) would occur in tandem to conserve PV. However, the lesson learned from the vorticity-and-stability diagram is that it seems as if the large-scale divergence drives a vorticity reduction, but the diabatic and adiabatic influences on static stability are engaged in a tug-of-war, such that PV is not conserved.

It could be that there is a gravity-wave driven hydrostatic-geostrophic adjustment tendency in response to latent heating (Blumen, 1972; Chagnon and Bannon, 2001; Egger, 2009) that drives the vorticity change. In the real-case simulation, the
WCB's PV change occurs also at a rate of 0.5$f$ per hour during the strong heating period, followed by a weaker decrease at 0.2$f$ per hour when heating becomes weak. But PV turns negative in the WCB outflow, which is not the desired outcome of geostrophic adjustment. Thus, future studies could explore the full vorticity budget (e.g., divergence, tilting and solenoid change under heating) and stability budget along the WCB flow. Such an analysis is beyond this study.

In a last case study, parcel trajectories that impinge on a warm front and are diabatically modified by parameterized con-
vection experience strong stability-driven PV modification. However, during the time of strongest convection, the temporal PV evolution is not well approximated by $\mathrm{PV}_{\mathrm{ver}}$, which limits what can be concluded from the two-dimensional vorticity-and-



stability diagram. Future work could also expand the diagram into a higher-dimensional phase space to account for changes in the horizontal components of vorticity and potential temperature gradients that form PV. Such a phase space could then be used to identify pathways of PV change common for different synoptic situations.

*Author contributions.* GV performed the analysis as part of his bachelor thesis supervised by ES and SeS. GV also prepared the manuscript. ES performed the IFS simulations, the corresponding trajectory computations and helped with the interpretation of the results. SeS conceived the idea for the vorticity-and-stability diagram and contributed to the interpretation and the paper writing.

*Competing interests.* The authors declare no competing interests.





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
