# Peer review of "A vorticity-and-stability diagram as a means to study potential vorticity nonconservation"

_Weather and Climate Dynamics, 2021_

## Referee Comment (RC2)

[referee-annotated manuscript omitted]

---

## Author Comment (AC1)

**Reply to the Reviewers' comments**

**General**

First, we would like to thank both reviewers for their thoughtful comments. At the heart of both reviews is the difficulty of separating PV conserving from PV non-conserving stability and vorticity changes in the diagram. It is argued that this information is not immediately apparent from the diagram. Further, both reviewers point out that PV conserving and non-conserving stability and vorticity changes can occur simultaneously, which is also not distinguishable in the diagram.

We argue that all the information requested by the reviewers is implicitly contained in the diagram and the separation into PV conserving and non-conserving stability and vorticity changes becomes apparent with only some minor modifications presented on the next pages.

Further, we argue that it is even possible to quantify the relative contributions by non-conserving stability and non-conserving vorticity changes to PV non-conservation without the need for diabatic model output. This is a key advantage of the diagram. Our motivation to study the PV composition and adiabatic and diabatic changes will now be presented at the end of this reply document. We hypothesize that the PV composition, i.e., its partioning into stability and vorticity, has important consequences for the downstream development. The diagram can help understand the adiabatic and diabatic contributions to the formation of a specific PV composition and its consequences for the downstream development.

On the next pages we outline how we suggest to adapt the diagram to make the requested information more accessible. Much of these explanations were missing in our original manuscript but the required changes are straightforward. Afterward, we also provide a point-by-point reply.

Kind regards The authors

**Suggested revisions**

**A vorticity-and-stability diagram to study the temporal evolution of the PV composition:**

We are interested in the nature of a change in the PV composition (in terms of its stability and vorticity). In the illustrative example in Fig.1, PV is changing between two times from 0.5 PVU to 1 PVU (black vector). The accompanying changes in stability and vorticity can be decomposed into PV conserving changes of stability and vorticity (change along the red hyperbola) and PV non-conserving changes of stability and vorticity (change along the gray hyperbola). In this example, obviously both contribute to the change in the PV composition.

Without loss of generality, we can define a rotated coordinate system (green coordinate system in Fig. 1) that is locally tangential to conserving stability and vorticity changes and with its second axes orthogonal to it, that is, the second axis is tangential to PV non-conserving stability and vorticity changes (the gray hyperboles). This procedure is standard and similar to what is done in differential calculus to estimate the derivative of a non-linear function using the slope of a line tangent to a point.

To quantify contributions by conservative and non-conserving stability and vorticity changes, which is requested by both reviews, the scalar products between the vector pointing into the direction of the change in the PV composition (black vector) and each of two base vectors of the local coordinate system (green vectors) is computed.

Figure 1: A change in the PV composition between two time steps ( $t_0$  and  $t_n$ ) can be decomposed into a change from PV conservative stability and vorticity changes (red) and PV non-conservative stability and vorticity changes (gray). To this end, a local coordinate system is defined (green) that is tangential to the conservative and non-conservative change (green). The

scalar products between the change vector (black) and the base vectors of the local coordinate system (green) quantify the relative contributions by PV conserving and non-conserving stability and vorticity changes. The scalar products between the vector tangent to the non-conservative change (gray) and the two base vectors of the stability-and-vorticity diagram (pink vectors lower left corner) quantify the relative importance of the PV non-conserving stability change compared with the PV non-conserving vorticity change. Axes are unitless. The red and gray vectors are curved only for illustration.

The nature of the decomposed change is now visualized in the local coordinate system (green in Fig. 1). If the vector in the local coordinate system in Fig. 2 points

- upward, it indicates a stability increase and vorticity decrease, which conserves PV;
- downward, it indicates a stability decrease and vorticity increase, which conserves PV;
- to the right, it indicates a PV increase;
- to the left, it indicates a PV decrease.

For the PV non-conservative change (vector points to the right or to the left), the relative importance of non-conservative stability compared to non-conservative vorticity change is indicated by the color of the dot (Fig. 2). To obtain it, we compute the scalar products between a vector tangent to the direction of PV non-conservation (gray hyperbolas in Fig. 1) with the two base vectors of the coordinate system of the original diagram (pink vectors in lower left corner in Fig. 1). Subsequently we compute the fraction of the two values. A low value (blue color) indicates the dominance of non-conserving stability change over non-conserving vorticity change; a high value (red color) indicates the dominance of non-conserving vorticity change.

As pointed out by the reviewers, simultaneous changes are common. If the vector in local coordinates, as in this example, points to the upper right in an approximate angle of  $45^{\circ}$  it indicates that adiabatic and diabatic stability and vorticity changes occur simultaneously and both change the PV composition equally. The length of the vector indicates the magnitude of the change. We also see from the color coding that for the non-conservative change (i.e., the PV increase) the non-conservative stability change (hence a diabatic stability increase) dominates over non-conservative vorticity change (blue dot in Fig. 2).